# DAWN-SI: Data-aware and Noise-Informed Stochastic Interpolation for Solving Inverse Problems

## Abstract

Inverse problems, which involve estimating parameters from incomplete or noisy observations, arise in various fields such as medical imaging, geophysics, and signal processing. These problems are often ill-posed, requiring regularization techniques to stabilize the solution. In this work, we employ *Stochastic Interpolation (SI)*, a generative framework that integrates both deterministic and stochastic processes to map a simple reference distribution, such as a Gaussian, to the target distribution. Our method **DAWN-SI**: **D**ata-**AW**are and **N**oise-informed **S**tochastic **I**nterpolation incorporates *data and noise embedding*, allowing the model to access representations about the measured data explicitly and also account for noise in the observations, making it particularly robust in scenarios where data is noisy or incomplete. By learning a time-dependent velocity field, SI not only provides accurate solutions but also enables uncertainty quantification by generating multiple plausible outcomes. Unlike pre-trained diffusion models, which may struggle in highly ill-posed settings, our approach is trained specifically for each inverse problem and adapts to varying noise levels. We validate the effectiveness and robustness of our method through extensive numerical experiments on tasks such as image deblurring and tomography.

## 1 Introduction

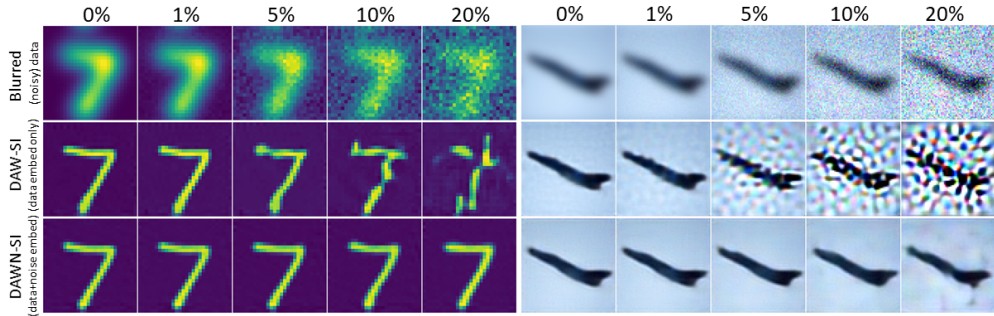

Figure 1: Example images from MNIST (left) and STL10 (right) datasets showing the recovery of deblurred images from the blurred (noisy) data (top panel) at different levels of noise (0%, 1%, 5%, 10%, 20%). Our methods DAW-SI (middle panel) incorporates the embedding for the blurred data within the network, while DAWN-SI (bottom panel) incorporates embedding for both data and noise levels. DAWN-SI is superior to DAW-SI at deblurred image recovery, especially at higher levels of noise.

Inverse problems are a class of problems in which the goal is to determine parameters (or parameter function) of a system from observed data. These problems arise in various fields, including medical imaging, geophysics, remote sensing, and signal processing. Inverse problems are often ill-posed, meaning that a unique solution does not exist, or the solution may be highly sensitive to small perturbations in the data. Solving inverse problems typically requires regularization techniques, which

introduce additional constraints or prior information to stabilize the solution and mitigate the effects of ill-posedness. Such regularization can be obtained using various mathematical and computational methods, including optimization techniques (variational methods), statistical inference (Bayesian or frequentist) and machine learning (Scherzer et al., 2009; Adler and Öktem, 2017; Calvetti and Somersalo, 2024; López-Tapia et al., 2021). In this paper, we approach inverse problems using the latter and investigate a new set of methods that are machine learning based for the solution of inverse problems. In particular, we show how stochastic interpolation (SI) which was recently proposed in the context of generative models can be effectively used to estimate the solution of inverse problems and to further investigate its non-uniqueness.

Stochastic Interpolation is a relatively new generative process that provides a unifying framework, elegantly integrating both deterministic flows and stochastic diffusion models (Albergo et al., 2023). The core concept of SI is to learn a stochastic process that effectively transports a simple reference distribution, such as Gaussian, to the desired target data distribution. This transportation process can manifest as either deterministic or stochastic. In the former case, it is described by an ordinary differential equation (ODE), while in the latter, it is governed by a stochastic differential equation (SDE).

The stochastic interpolation framework defines a continuous-time reversibility between the reference and target distributions, parameterized by time $t \in [0, 1]$. At the initial time $t = 0$, the distribution aligns with the reference distribution. As time progresses to $t = 1$, the distribution evolves to match the target data distribution. This evolution is achieved by learning the time-dependent velocity field (for ODEs) or drift and diffusion coefficients (for SDEs) that characterize this interpolation process. By understanding and modeling this time-dependent transformation, one can generate samples from the target distribution through numerical integration of the learned ODE or SDE.

SI is a highly flexible methodology for designing new types of generative models. In this work, we use this flexibility and show how to adopt SI to solve highly ill-posed inverse problems. We find SI particularly useful since it allows ease of sampling from the target distribution. This implies that we are able to generate a range of solutions to the inverse problems and thus investigate the uncertainty that is associated with the estimated solution. Such a process is highly important in physical applications when decisions are made based on the solution.

**Related work:**  The methods proposed here are closely related to three different approaches for the solution of inverse problems. First, there is an obvious link to the incorporation of diffusion models as regularizers in inverse problems (Yang et al., 2022; Chung et al., 2022a;c;b; Song et al., 2022). The key idea is to leverage a pre-trained diffusion model that captures the data distribution as a prior, and then condition this model on the given measurements to infer the underlying clean signal or image. For inverse problems, diffusion methods condition the diffusion model on the given measurements (e.g. noisy, incomplete, or compressed data) by incorporating them into the denoising process. Nonetheless, it has been shown in Eliasof et al. (2024) that pre-trained diffusion models that are used for ill-posed inverse problems as regularizers tend to under-perform as compared to the models that are trained specifically on a particular inverse problem. In particular, such models tend to break when the noise level is not very low.

A second branch of techniques that are related to the work proposed here use encoder-decoder type networks for the solution of inverse problems (see Chung et al. (2024); Thomas et al. (2022) and reference within). Such approaches are sometimes referred to as likelihood-free estimators as they yield a solution without the computation of the likelihood. This technique is particularly useful for problems where the forward problem is difficult to compute.

A third branch of techniques that relates to our approach uses the forward problem within the neural network (see Eliasof et al. (2024); Mukherjee et al. (2021); Eliasof et al. (2023); Jin et al. (2017) and reference within). In this approach, one computes the data misfit (and its gradient) within the network and use it to guide training. Our approach utilizes components from this methodology to deal with the measured data.

**Main Contribution:**  The core contributions of this paper lie in the design and application of the DAWN-SI framework, specifically crafted for solving highly ill-posed inverse problems like image deblurring or tomography. (i) Our method is designed to be problem-specific, which adjusts itself to the unique structure of the inverse problem it is tasked with, learning the posterior distribution

directly, ensuring that the learned velocity fields and mappings are directly applicable to the target task. (ii) We train a stochastic interpolant by embedding measured data and noise information directly into the interpolation process. This incorporation allows our model to adapt to a wide range of noise conditions, something that pre-trained models struggle with. By making the training explicitly aware of the noise level and data characteristics, our model can better navigate noisy or incomplete measured data, producing superior reconstructions. (iii) By learning the posterior distribution directly and leveraging the stochastic nature of the interpolation process, DAWN-SI generates multiple plausible solutions for a given inverse problem, allowing us to explore the solution space more thoroughly and in particular, to estimate the posterior mean and its standard deviation, estimating the uncertainty in the recovered solution.

## 2 STOCHASTIC INTERPOLATION AND INVERSE PROBLEMS

In this section we review stochastic interpolation as well as derive the main ideas behind using it for the solution of inverse problems.

### 2.1 STOCHASTIC INTERPOLATION: A PARTIAL REVIEW

Stochastic interpolation (SI) is a framework that transforms points between two distributions. Given two densities $\pi_0(\mathbf{x})$ and $\pi_1(\mathbf{x})$, the goal is to find a mapping that takes a point $\mathbf{x}_0 \sim \pi_0(\mathbf{x})$ and transports it to a point $\mathbf{x}_1 \sim \pi_1(\mathbf{x})$. For simplicity and for the purpose of this work, we choose $\pi_0$ to be a Gaussian distribution with $0$ mean and $\mathbf{I}$ covariance.

Consider sampling points from both distributions and define the trajectories

$$\mathbf{x}_t = (1-t)\mathbf{x}_0 + t\mathbf{x}_1 \tag{1}$$

These trajectories connect points from $\mathbf{x}_0$ at $t = 0$ to $\mathbf{x}_1$ at $t = 1$. More complex trajectories have been proposed in Albergo et al. (2023), however, in our context, we found that simple linear trajectories suffice and have advantages for being very smooth in time. The velocity along the trajectory is the time derivative of the trajectory, that is,

$$\frac{d\mathbf{x}_t}{dt} = \mathbf{v} = \mathbf{x}_1 - \mathbf{x}_0 \tag{2}$$

Under the SI framework, the velocity field for all $(\mathbf{x}_t, t)$ is learned by averaging over all possible paths. To this end, we parameterize the velocity by a function $\mathbf{s}_\theta(\mathbf{x}_t, t)$ and solve the stochastic optimization problem for $\boldsymbol{\theta}$

$$\widehat{\boldsymbol{\theta}} = \arg\min_{\boldsymbol{\theta}} \mathbb{E}_{\mathbf{x}_0, \mathbf{x}_1} \left[ \|\mathbf{s}_\theta(\mathbf{x}_t, t) - \mathbf{v}\|^2 \right] = \arg\min_{\boldsymbol{\theta}} \mathbb{E}_{\mathbf{x}_0, \mathbf{x}_1} \left[ \|\mathbf{s}_\theta(\mathbf{x}_t, t) + \mathbf{x}_0 - \mathbf{x}_1\|^2 \right] \tag{3}$$

After training, the velocity function $\mathbf{s}_\theta(\mathbf{x}_t, t)$ can be estimated for every point in time. In this work, given $\mathbf{x}_0$, we recover $\mathbf{x}_1$ by numerically integrating the ODE

$$\frac{d\mathbf{x}_t}{dt} = \mathbf{s}_\theta(\mathbf{x}_t, t), \quad \mathbf{x}(0) = \mathbf{x}_0, \quad t \in [0, 1], \tag{4}$$

by Fourth-Order Runge Kutta method with fixed step size. An alternative version where the samples are obtained by using a stochastic differential equation can also be used.

While it is possible to estimate the velocity $\mathbf{v}$ from $(\mathbf{x}_t, t)$ and then compare it to the true velocity, it is sometimes easier to work with the velocity as a denoiser network, that is, estimate $\mathbf{x}_1$ and use the loss of comparing $\mathbf{x}_1$ to its denoised quantity. To this end, note that

$$\mathbf{x}_1 = \mathbf{x}_t + (1-t)\mathbf{v} \approx \mathbf{x}_t + (1-t)\mathbf{s}_\theta(\mathbf{x}_t, t) \tag{5}$$

Equation (5) is useful when a recovered $\mathbf{x}_1$ is desirable. In the context of using SI for inverse problems, obtaining an approximation to $\mathbf{x}_1$ can be desirable, as we see next.

## 2.2 APPLYING STOCHASTIC INTERPOLATION FOR THE SOLUTION OF INVERSE PROBLEMS

Consider the case where we have observations on $\mathbf{x}_1$ of the form

$$\mathbf{A}\mathbf{x}_1 + \boldsymbol{\varepsilon} = \mathbf{b} \tag{6}$$

Here, $\mathbf{A}$ is a linear forward mapping (although the method can work for nonlinear mappings as well) and $\boldsymbol{\varepsilon} \sim \mathcal{N}(0, \sigma^2\mathbf{I})$ is a random vector. We assume that $\mathbf{A}$ is rank-deficient or numerically rank-deficient (Hansen, 1997), so the effective dimension of $\mathbf{b}$ is smaller than $\mathbf{x}_1$ and one cannot obtain a reasonable estimate for $\mathbf{x}_1$ given the noisy data $\mathbf{b}$ without the incorporation of a-priori information.

Using Bayes' theorem, we have

$$\pi(\mathbf{x}_1|\mathbf{b}) \propto \pi(\mathbf{b}|\mathbf{x}_1)\pi(\mathbf{x}_1) \tag{7}$$

Bayes' theorem suggests that it is possible to factor the posterior distribution $\pi(\mathbf{x}_1|\mathbf{b})$ using the known distribution of $\pi(\mathbf{b}|\mathbf{x}_1)$ and the prior distribution $\pi(\mathbf{x}_1)$. This observation motivated a number of studies that used the estimated pre-trained distribution $\pi(\mathbf{x}_1)$ in the process of solving an inverse problem (Yang et al., 2022; Chung et al., 2022a;c;b; Song et al., 2022). Nonetheless, it has been shown in Eliasof et al. (2024) that such estimators tend to produce unsatisfactory results when solving highly ill-posed problems or when the data is very noisy. The reason for this behaviour stems from the fact that pre-trained estimators push the solution towards the center of the prior distribution, irrespective of what the data represents. To see this, we use a careful dissection of the solution. Consider the singular value decomposition

$$\mathbf{A} = \sum_i \boldsymbol{\lambda}_i \mathbf{u}_i \mathbf{v}_i^\top$$

where $\mathbf{u}_i$ and $\mathbf{v}_i$ are the left and right singular vectors and $\boldsymbol{\lambda}_i$ are the singular values. Given the orthogonality of $\mathbf{u}$, we can decouple the data equations into

$$\boldsymbol{\lambda}_i(\mathbf{v}_i^\top \mathbf{x}_1) = (\mathbf{u}_i^\top \mathbf{b}), \quad i = 1, \ldots, n$$

If $\boldsymbol{\lambda}_i$ is large, the projection of $\mathbf{x}_1$ onto the eigenvector $\mathbf{v}_i$ is very informative and very minimal regularization is required. However, if $\boldsymbol{\lambda}_i \approx 0$, the contribution of $\mathbf{v}_i$ to the solution is difficult, if not impossible, to obtain and this is where the regularization is highly needed. When using pre-trained models, the prior $\pi(\mathbf{x}_1)$ is estimated numerically and it is unaware of the inverse problem at hand. Errors in the estimated $\pi(\mathbf{x}_1)$ in the parts that correspond to the large singular vectors may not be destructive. However, if $\pi(\mathbf{x}_1)$ has errors that correspond to the very small singular values, that is, to the effective null space of the data, this may lead to the artifacts that have been observed early in Kaipio and Somersalo (2004); Tenorio et al. (2011). This suggests that although it is appealing to use a generic pre-trained priors in the process, a better approach is to not use the Bayesian factorization to prior and likelihood but rather train a stochastic interpolant that maps the distribution $\pi(\mathbf{x}_0)$ to the posterior $\pi(\mathbf{x}_1|\mathbf{b})$ directly. Indeed, as we show next, the flexibility of the SI framework allows us to learn a velocity function that achieves just that.

## 2.3 A DATA-AWARE AND NOISE-INFORMED VELOCITY ESTIMATOR

In the canonical form of stochastic interpolation, the velocity is estimated from the interpolated vector $\mathbf{x}_t$. In the context of inverse problems, we also have a vector $\mathbf{b}$ (measured data) that contains additional information on $\mathbf{x}_1$ and therefore can be used to estimate the velocity towards the posterior. We now show that by doing a small change to the training process of SI, it is possible to solve inverse problems using the same concept.

To this end, notice that we train a velocity function $\mathbf{s}_{\boldsymbol{\theta}}(\mathbf{x}_t, t)$ that takes in two arguments. In the context of a specific inverse problem, we have additional information for training the network, the measured data $\mathbf{b}$ and the noise level $\sigma$ in $\mathbf{b}$. Note that it is also possible to estimate the noise level directly from the data as proposed in Tenorio et al. (2011). The data $\mathbf{b}$ can be used to point toward $\mathbf{x}_1$ even at time $t = 0$, where $\mathbf{x}_t$ contains no information on $\mathbf{x}_1$ and can therefore improve the estimation of velocity. Moreover, the information about noise level $\sigma$ in the measured data during training also makes the estimator more robust to noise during inference. We thus propose to use the data and noise when estimating $\mathbf{s}_{\boldsymbol{\theta}}$. To this end, we use a transformation $f$ of data $\mathbf{b}$ (to be discussed next), and let

$$\mathbf{s}_{\boldsymbol{\theta}} = \mathbf{s}_{\boldsymbol{\theta}}(\mathbf{x}_t, f(\mathbf{b}), t, \sigma) = \mathbf{s}_{\boldsymbol{\theta}}(\mathbf{x}_t, f(\mathbf{A}\mathbf{x}_1 + \sigma\mathbf{z}), t, \sigma), \tag{8}$$

where $\mathbf{z} \sim \mathcal{N}(0, \mathbf{I})$. To estimate $\boldsymbol{\theta}$, we simply repeat the minimization process as before and match the flow where the data is a part of the estimated velocity, that is,

$$\widehat{\boldsymbol{\theta}} = \arg\min_{\boldsymbol{\theta}} \mathcal{L}_1(\boldsymbol{\theta}) = \arg\min_{\boldsymbol{\theta}} \mathbb{E}_{\mathbf{x}_0, \mathbf{x}_1, \sigma, t} \left[ \|\mathbf{s}_\theta(\mathbf{x}_t, f(\mathbf{A}\mathbf{x}_1 + \sigma\mathbf{z}), t, \sigma) + \mathbf{x}_0 - \mathbf{x}_1\|^2 \right], \quad (9)$$

where $\mathcal{L}_1$ represents the mean loss in the prediction of velocity. The trained network can then be used to invert new data. Let us assume that we are given some fixed vector $\mathbf{b}$ and we want to estimate $\mathbf{x}_1$. This can be done simply by solving the ODE

$$\frac{d\mathbf{x}_t}{dt} = \mathbf{s}_{\boldsymbol{\theta}}(\mathbf{x}_t, f(\mathbf{b}), t, \sigma), \quad \mathbf{x}(0) = \mathbf{x}_0, \quad t \in [0, 1], \quad (10)$$

where $\mathbf{b}$ and $\sigma$ are now fixed.

As we show in our numerical experiments, having $\sigma$ as an input to the network plays an important role, generating an inversion methodology that is robust to different noise levels.

An important question is the design of a network that integrates the information about $\mathbf{b}$ into the velocity estimation process. One important choice is the function $f$ that operates on $\mathbf{b}$. For many, if not most, inverse problems, the data $\mathbf{b}$ belongs to a different space than $\mathbf{x}$. Therefore, it is difficult to use this vector directly. The goal of the function $f$ is to transform the data, $\mathbf{b}$ from the data space to the space of $\mathbf{x}$. One obvious approach to achieve this is to choose

$$f(\mathbf{b}) = \mathbf{A}^\top \mathbf{b}. \quad (11)$$

This approach was used in Mardani et al. (2018); Adler and Öktem (2017), and as shown in Section 2.5, can be successful for the transformation of the data into the image space. Other possible approaches can include fast estimation techniques for $\mathbf{x}$ given $\mathbf{b}$ such as the conjugate gradient least squares method (Hansen, 1997). For the experiments presented here, we found that using the adjoint $\mathbf{A}^\top$ of the forward problem was sufficient.

To demonstrate these points, we consider the following toy example.

**Example 2.1 The Duathlon problem:** *We consider the duathlon problem where one records the total time it takes to perform a duathlon (bike and run). Given the total time, the goal is to recover the time it takes to perform each individual segment. The problem is clearly under-determined as one has to recover two number given a single data point. Let $\mathbf{x} = [x_1, x_2]$ be the vector, where $x_1$ and $x_2$ represent the time it takes to finish the bike and run segments, respectively. The data is simply*

$$b = x_1 + x_2 + \epsilon$$

*Assume that we have a prior knowledge that the distribution of $\mathbf{x}$ is composed of two Gaussians. Using SI to generate data from these Gaussians is demonstrated in Figure 2.*

*The data is obtained by training a network, approximately solving the optimization problem in Equation (3) and using Equation (4) to integrate $\mathbf{x}_0$ that is randomly chosen from a Gaussian.*

*Now, in order to solve the inverse problem, we train a larger network that includes the data and approximately solves the optimization problem in Equation (9). Given the data $\mathbf{b}$, we now integrate the ODE Equation (10) to obtain a solution. The result of this integration is presented in Figure 2 (right). We observe that not only did the process identify the correct lobe of the distribution, it also sampled many solutions around it. This enables us in obtaining not just a single but rather an ensemble of plausible solutions that can aid in exploring uncertainty in the result.*

## 2.4 TRAINING THE SI MODEL FOR INVERSE PROBLEMS

When training an SI model, we solve the optimization problem given by Equation (3). In this process, one draws samples from $\mathbf{x}_0$ and $\mathbf{x}_1$, then randomly chooses $t \in [0, 1]$ to generate the vector $\mathbf{x}_t$. In addition, we generate the data $\mathbf{b}$ by multiplying the matrix $\mathbf{A}$ with $\mathbf{x}_1$ and adding noise $\epsilon$ with a random standard deviation $\sigma$.

Next, one feeds the network $\mathbf{x}_t$, $t$, $\mathbf{b}$ and $\sigma$ to compute $\mathbf{s}_\theta(\mathbf{x}_t, \mathbf{b}, t, \sigma)$ and then compare it to the velocity $\mathbf{v} = \mathbf{x}_1 - \mathbf{x}_0$.

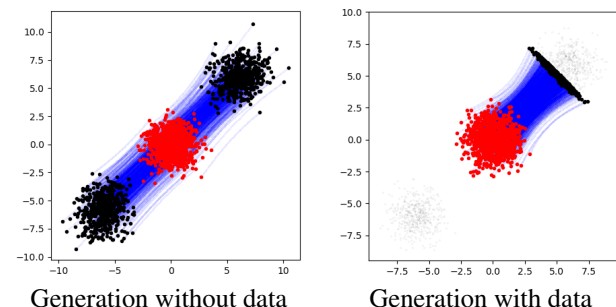

Generation without data       Generation with data

Figure 2: Generation of the distribution of two Gaussians by SI without and with data.

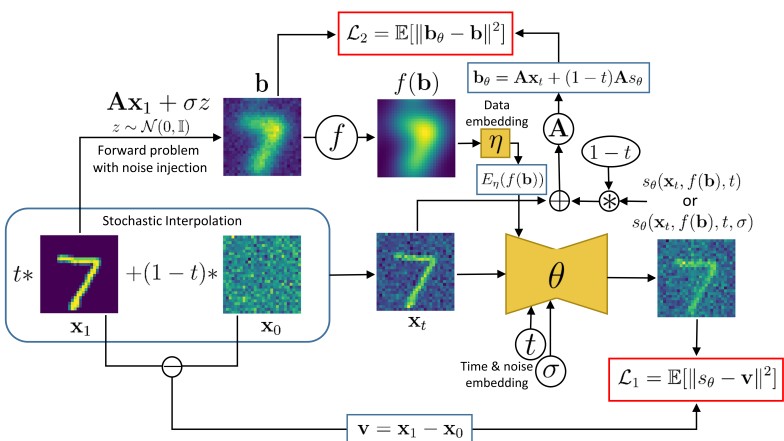

Figure 3: Schematic of the training process for the SI model for solving inverse problem, where the forward model is given by $\mathbf{A}$. This figure specifically represents the deblurring inverse problem. The network with parameters $\boldsymbol{\theta}$ can either have noise embedding (DAWN-SI) or not (DAW-SI). The two loss terms $\mathcal{L}_1$ and $\mathcal{L}_2$ represent the error in prediction of velocity and misfit, respectively. In this figure, the transformation $f$ for generating the data embedding was chosen as $f = \mathbf{A}^\top$.

While it is possible to train the network by comparing the computed velocity to its theoretical value, we have found that adding additional terms to the loss helps in getting a better estimate for the velocity. In particular, when training the model for inverse problems, we add an additional term to the loss that relates to the particular inverse problem in mind. To this end, note that after estimating $\mathbf{v}$, we can estimate $\mathbf{x}_1$ using Equation (5). Thus, we can estimate the data fit $\mathbf{b}_{\boldsymbol{\theta}}$ for the estimated velocity, given by,

$$\mathbf{b}_{\boldsymbol{\theta}} = \mathbf{A}\mathbf{x}_t + (1-t)\mathbf{A}\mathbf{s}_{\boldsymbol{\theta}}(\mathbf{x}_t, f(\mathbf{b}), t, \sigma) \tag{12}$$

If the velocity is estimated exactly, then $\mathbf{b}_{\boldsymbol{\theta}} = \mathbf{b}$. Therefore, a natural loss for the recovery is the comparison of the recovered data to the given data. This implies that the velocity function should honor the data $\mathbf{b}$, as well as the original distribution $\mathbf{x}_1$. We thus introduce the misfit loss term,

$$\mathcal{L}_2(\boldsymbol{\theta}) = \mathbb{E}_{\mathbf{x}_0, \mathbf{x}_1, \sigma, t}\left[\|\mathbf{b}_{\boldsymbol{\theta}} - \mathbf{b}\|^2\right], \tag{13}$$

that pushes the recovered $\mathbf{v}$ to generate an $\mathbf{x}_1$ that fits the measured data.

To summarize, we modify the training by solving the following optimization problem,

$$\widehat{\boldsymbol{\theta}} = \arg\min_{\boldsymbol{\theta}}\left\{\mathbb{E}_{\mathbf{x}_0, \mathbf{x}_1, \sigma, t}\|\mathbf{s}_{\boldsymbol{\theta}}(\mathbf{x}_t, f(\mathbf{b}), t, \sigma) - \mathbf{v}\|^2 + \alpha\mathbb{E}_{\mathbf{x}_0, \mathbf{x}_1, \sigma, t}\|\mathbf{b}_{\boldsymbol{\theta}} - \mathbf{b}\|^2\right\}, \tag{14}$$

where $\alpha$ is a hyperparameter, which we set to 1. A schematic for training of SI model is given in Figure 3.

### 2.5 Architectures for data-aware and noise-informed velocity estimators

Our goal is to train a velocity estimator $s_\theta(x_t, b, t, \sigma)$ and then compare it to the velocity $v$. Our estimator is based on a UNet (Ronneberger et al., 2015) with particular embeddings for $b$, $t$, and $\sigma$.

The embedding of time and noise is straight-forward. Note that both $t$ and $\sigma$ are scalars. For their embedding, we use the method presented in Croitoru et al. (2023), which involves creating learnable embedding and adding them to the feature maps at each level of the network.

A key component in making SI perform well for inverse problems is the careful design of a network that incorporates the data within the training process. The resulting network can be thought of as a likelihood-free estimator (Thomas et al., 2022), that estimates the velocity vector $v$, given the input vector $x_t$, the time $t$ and the noise level $\sigma$. As previously explained, we do not integrate $b$ directly, but rather use $f(b) = A^\top b$ and embed this vector in the network.

The embedding of $A^\top b$ is performed using a data encoder network. To this end, let

$$E_\eta(A^\top b) \tag{15}$$

be an encoder network that is parameterized by $\eta$. The encoder can be pre-trained or trained as part of the network, providing flexibility in how it is integrated into the UNet. The encoded data provides additional context that is critical for accurate predictions. Formally, the UNet estimates the velocity as follows:

$$\widehat{v} = s_\theta(x_t, E_\eta(A^\top b), t, \sigma) \tag{16}$$

By embedding both time, data and noise vectors at each layer, the network can leverage additional information, leading to more accurate and robust predictions, especially for large noise levels.

### 2.6 Inference and Uncertainty Estimation

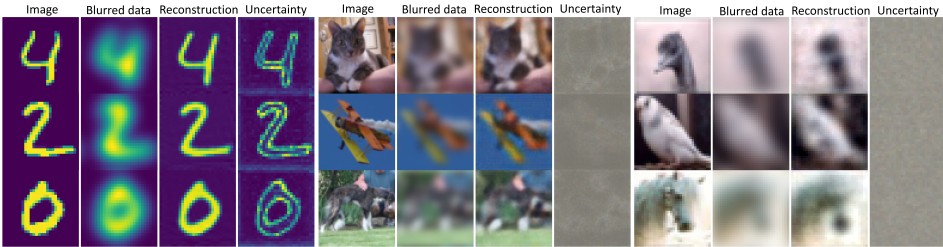

Figure 4: Computing uncertainty in the solutions obtained by SI for some example images from MNIST (left), STL10 (middle) and CIFAR10 (right) for the deblurring task. The reconstruction (posterior mean) is computed by averaging over runs from 32 randomly initialized $x_0$. The uncertainty in the posterior mean is computed by evaluating its standard deviation for the predictions from 32 runs.

Our method is specifically aimed at highly ill-posed inverse problems, where regularization is often necessary to arrive at a stable solution. These problems often do not have a unique solution, and slight changes in input data can lead to large variations in the solution. Due to its stochastic nature, our method allows for the realization of multiple solutions since one can start with many random initial points $x_0$ and evolve to multiple versions $x_1$ of the solutions. This property can be used to generate even better estimates and for the estimation of uncertainty in the recovered images. To this end, assume that the ODE is solved $M$ times starting each time at a different initial condition $x_0^{(j)}$ and let

$$x_1^{(j)} = x_t(t = 1, x_0 = x_0^{(j)}), \quad j = 1, \dots, M \tag{17}$$

be the solution of the ODE in Equation (9) obtained at time $t = 1$ starting from point $x_0^{(j)}$ at $t = 0$. The points $x_1^{(j)}$ for $j = 1, \dots, M$ represent an ensemble of solutions, that are sampled from the posterior $\pi(x_1 | b)$. Given these points, it is simple to estimate the mean and standard deviation of the posterior distribution. In particular, we define

$$\bar{x}_1 = \frac{1}{M} \sum_{j=1}^{M} x_1^{(j)}, \quad \sigma_{\bar{x}_1} = \frac{1}{M} \sum_{j=1}^{M} \|x_1^{(j)} - \bar{x}_1\|^2 \tag{18}$$

as the mean and standard deviation of the estimated solutions. In particular, $\bar{\mathbf{x}}_1$ approximates the posterior mean and $\sigma_{\bar{\mathbf{x}}_1}$ estimates its standard deviation. It is well known that, for many problems the posterior mean can have a lower risk compared to other estimators (e.g. the Maximum A-Posteriori estimator). For an elaborate discussion on the properties of such estimators, see Tenorio et al. (2011); Kaipio and Somersalo (2004); Calvetti and Somersalo (2005). While such estimators are typically avoided due to computational complexity, the computational framework presented here allows us to compute them relatively easily.

An example for this process is shown in Figure 4. Here, the solutions over multiple runs were averaged to generate an effective reconstruction of the original image (the posterior mean). Moreover, to quantify uncertainty in the reconstruction process, we computed standard deviation over the solutions at each pixel from multiple runs. In the uncertainty maps for MNIST, the uncertainty is concentrated along the edges of the digits. This occurs because the SI model introduces slight variations in how it reconstructs the boundary between the digit and the background. Since this boundary is sharp, any slight differences in how this edge is defined in different reconstructions lead to higher uncertainty along the edges. On the other hand, for STL10 and CIFAR10 images, the boundary between objects and background is often less distinct. The background might contain detailed textures or noise that blends into the object, making it harder for the model to distinguish clear boundaries. Hence, the uncertainty maps for these datasets do not exhibit the same clear edge-focused uncertainty as in MNIST. The lack of a clear boundary means that the reconstruction's variability spreads more evenly across the entire image.

## 3 NUMERICAL EXPERIMENTS

In this section, we experiment with our method on a few common datasets and two broadly applicable inverse problems: image deblurring and tomography (see Appendix A for details). We provide additional information on the experimental settings, hyperparameter choices, and network architectures in Tables 3 to 5 for image deblurring task and Table 6 for tomography task in Appendix C.

**Training methodology.** For our experiments, we considered two types of data-aware velocity estimator networks: (i) DAW-SI: the estimator with no noise-embedding, and (ii) DAWN-SI: the estimator with trainable noise-embedding. We employed antithetic sampling during training. Starting with an input batch of clean images $\mathbf{x}_1'$, an $\mathbf{x}_0' \sim \mathcal{N}(0, \mathbf{I})$ was sampled and antithetic pairs $\mathbf{x}_1$ and $\mathbf{x}_0$,

$$\mathbf{x}_1 = \begin{bmatrix} \mathbf{x}_1' \\ \mathbf{x}_1' \end{bmatrix}, \quad \mathbf{x}_0 = \begin{bmatrix} \mathbf{x}_0' \\ -\mathbf{x}_0' \end{bmatrix} \tag{19}$$

were generated by concatenation along the batch dimension. For a large number of samples, the sample mean of independent random variables converges to the true mean. However, the convergence can be slow due to the variance of the estimator being large. Antithetic sampling helped reduce this variance by generating pairs of negatively correlated samples, thereby improving convergence. The data was generated by employing the forward model along with a Gaussian noise injection, $\mathbf{b} = \mathbf{A}\mathbf{x}_1 + \sigma \mathbf{z}$, where $\mathbf{z} \sim \mathcal{N}(0, \mathbf{I})$. The value of $\sigma$ was set to $p\%$ of the range of values in the data $\mathbf{b}$, where $p$ was sampled uniformly in $(0, 20)$. $\mathbf{x}_t$ and $\mathbf{v}$ were computed from $\mathbf{x}_1$ and $\mathbf{x}_0$ following Equation (1) and Equation (2). Using the transformation $E_{\boldsymbol{\eta}}(f(\mathbf{b})) = E_{\boldsymbol{\eta}}(\mathbf{A}^\top \mathbf{b})$, the predicted velocity of the estimator was $s_{\boldsymbol{\theta}}(\mathbf{x}_t, E_{\boldsymbol{\eta}}(\mathbf{A}^\top \mathbf{b}), t)$ for the DAW-SI estimator and $s_{\boldsymbol{\theta}}(\mathbf{x}_t, E_{\boldsymbol{\eta}}(\mathbf{A}^\top \mathbf{b}), t, \sigma)$ for the DAWN-SI estimator. Here, $E_{\boldsymbol{\eta}}$ was chosen to be a single-layer convolutional neural network. The loss for an epoch was computed as given in Equation (14).

**Inference**. For inference on a noisy data, $\mathbf{b} = \mathbf{A}\mathbf{x}_1 + \sigma \mathbf{z}$ for some image $\mathbf{x}_1$, we start with a randomly sampled $\mathbf{x}_0 \sim \mathcal{N}(0, \mathbf{I})$ at $t = 0$ and perform Fourth-Order Runga Kutta numerical integration to solve the ODE in Equation (10) with a step size $h = 1/100$, where the velocity was computed using the trained estimator as $s_{\boldsymbol{\theta}}(\mathbf{x}_t, E_{\boldsymbol{\eta}}(\mathbf{A}^\top \mathbf{b}), t)$ for the DAW-SI estimator and $s_{\boldsymbol{\theta}}(\mathbf{x}_t, E_{\boldsymbol{\eta}}(\mathbf{A}^\top \mathbf{b}), t, \sigma)$ for the DAWN-SI estimator at time $t$. For each image $\mathbf{x}_1$, the inference was run 32 times starting from 32 different realizations of $\mathbf{x}_0$ and the resulting reconstructed images (which are samples from the posterior $\pi(\mathbf{x}_1|\mathbf{b})$) were averaged to generate the final recovered image. The uncertainty at the pixel-level could also be estimated from these samples by computing their standard deviation, as shown in Figure 4.

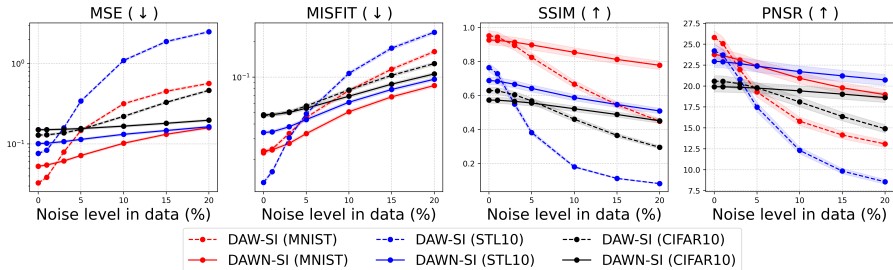

Figure 5: Assessing the sensitivity of the metrics for the recovered deblurred image as a function of the level of noise ($p\%$) added to the data. The DAWN-SI is more robust to noise across all three datasets, MNIST, STL10 and CIFAR10 than DAW-SI, with the crossing point happening at values of $p \lesssim 5\%$ for all metrics.

Table 1: Comparison between the performances of different methods (Diffusion, InverseUNetODE, and our methods DAW-SI and DAWN-SI) for the deblurring task for MNIST, STL10 and CIFAR10 datasets using metrics MSE, MISFIT, SSIM and PSNR. All the evaluations have been presented on the test set for a large blurring kernel (for details, see Appendix A.1) at a noise level $p = 5\%$ in the blurred data. Here, we report mean $\pm$ standard deviation over all images in the test set. First and second best performances are shown in **bold** and red, respectively.

| Dataset | Methods | MSE ($\downarrow$) | MISFIT ($\downarrow$) | SSIM ($\uparrow$) | PSNR ($\uparrow$) |
|---|---|---|---|---|---|
| MNIST | Diffusion | $6.078 \pm 8.508$ | $0.045 \pm 0.008$ | $0.248 \pm 0.178$ | $6.538 \pm 6.980$ |
| | InverseUNetODE | $0.167 \pm 0.072$ | $0.055 \pm 0.012$ | $0.714 \pm 0.090$ | $18.394 \pm 1.946$ |
| | DAW-SI (Ours) | $0.156 \pm 0.078$ | $0.043 \pm 0.012$ | $0.825 \pm 0.062$ | $18.899 \pm 2.472$ |
| | DAWN-SI (Ours) | $\mathbf{0.073 \pm 0.041}$ | $\mathbf{0.032 \pm 0.009}$ | $\mathbf{0.901 \pm 0.042}$ | $\mathbf{22.277 \pm 2.612}$ |
| STL10 | Diffusion | $3.712 \pm 3.900$ | $0.050 \pm 0.012$ | $0.170 \pm 0.057$ | $8.581 \pm 3.765$ |
| | InverseUNetODE | $0.299 \pm 0.110$ | $0.084 \pm 0.063$ | $0.360 \pm 0.059$ | $18.427 \pm 1.342$ |
| | DAW-SI (Ours) | $0.344 \pm 0.132$ | $0.049 \pm 0.012$ | $0.382 \pm 0.064$ | $17.434 \pm 1.465$ |
| | DAWN-SI (Ours) | $\mathbf{0.113 \pm 0.056}$ | $\mathbf{0.043 \pm 0.013}$ | $\mathbf{0.644 \pm 0.082}$ | $\mathbf{22.423 \pm 1.904}$ |
| CIFAR10 | Diffusion | $1.049 \pm 0.749$ | $\mathbf{0.042 \pm 0.012}$ | $0.272 \pm 0.081$ | $12.104 \pm 2.888$ |
| | InverseUNetODE | $0.168 \pm 0.080$ | $0.067 \pm 0.027$ | $0.544 \pm 0.075$ | $19.388 \pm 1.587$ |
| | DAW-SI (Ours) | $\mathbf{0.152 \pm 0.071}$ | $0.057 \pm 0.025$ | $\mathbf{0.567 \pm 0.077}$ | $\mathbf{19.765 \pm 1.520}$ |
| | DAWN-SI (Ours) | $0.156 \pm 0.077$ | $0.055 \pm 0.023$ | $0.554 \pm 0.081$ | $19.743 \pm 1.712$ |

**Baselines.** For the image deblurring task, we compare our methods DAW-SI and DAWN-SI to diffusion-based image deblurring (Chung et al., 2022a) and InverseUNetODE (Eliasof et al., 2024). The details for the training setup and hyperparameter choices for the baselines have been provided in Appendix C. All baselines were trained with Gaussian noise injection in the data in the same way as used for training of DAW-SI and DAWN-SI methods for fair comparison.

**Metrics.** For performance evaluation, we compute mean squared error (MSE), misfit, structural similarity index measure (SSIM) and peak signal-to-noise ratio (PSNR) between the ground truth and the reconstructed image. Additional details on these metrics are provided in Appendix B.

### 3.1 IMAGE DEBLURRING

We present the results of image deblurring task in Table 1 at noise level $p = 5\%$ in blurred data for MNIST, STL10 and CIFAR10 datasets. For all metrics, DAWN-SI beats all other methods for all datasets by large margins, except for the CIFAR10 dataset, where DAW-SI performed marginally better than DAWN-SI. Moreover, on the CIFAR10 dataset, the Diffusion model fits the data the best achieving the lowest value of the misfit metric. We also performed ablation studies for different noise levels in data for both DAW-SI and DAWN-SI models, as shown in Figure 5. For all metrics, DAWN-SI was more robust to noise in data, especially for higher noise levels over all datasets with the crossing point happening at values of $p \lesssim 5\%$ for all metrics. Some example images for the deblurring task for different levels of noise in data are presented in Figures 7 to 9 in Appendix D. Based on empirical evidence from training, we believe that the results for both DAW-SI and DAWN-SI could be improved further with more training epochs.

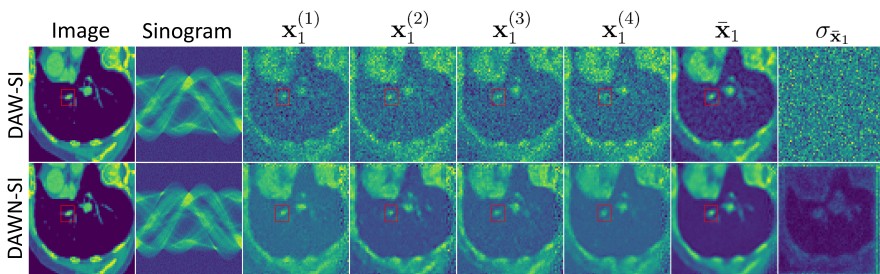

Figure 6: An example from OrganAMNIST dataset showing the different samples (labeled $\mathbf{x}_1^{(1)}$ to $\mathbf{x}_1^{(4)}$) drawn from the posterior starting from different initial $\mathbf{x}_0$ for both DAW-SI (top) and DAWN-SI (bottom). The posterior mean $\bar{\mathbf{x}}_1$ and standard deviation $\sigma_{\bar{\mathbf{x}}_1}$ were computed from 32 such samples, given the sinogram data (at noise level $p = 5\%$). The red box highlights a lobe, which exhibits variability in its size and shape across different samples. This variability reflects the model's uncertainty in reconstructing (the boundary of) such anatomical features.

Table 2: Comparison between the performances of our methods (DAW-SI and DAWN-SI) for the tomography task for OrganAMNIST and OrganCMNIST datasets using metrics MSE, MISFIT, SSIM and PSNR. All the evaluations have been presented on the test set for $N_{\text{angles}} = 360$, and $N_{\text{detectors}} = 2s + 1$, where $s \times s$ is the image dimension (for details, see Appendix A.2) at a noise level $p = 5\%$ in the sinogram data. Here, we report mean $\pm$ standard deviation over 5000 images in the test set. Best performance is shown in **bold**.

| Dataset | Methods | MSE ($\downarrow$) | MISFIT ($\downarrow$) | SSIM ($\uparrow$) | PSNR ($\uparrow$) |
|---|---|---|---|---|---|
| OrganAMNIST | DAW-SI (Ours) | $0.008 \pm 0.002$ | $0.041 \pm 0.051$ | $0.575 \pm 0.108$ | $20.389 \pm 1.767$ |
| | DAWN-SI (Ours) | $\mathbf{0.004 \pm 0.001}$ | $\mathbf{0.036 \pm 0.060}$ | $\mathbf{0.713 \pm 0.090}$ | $\mathbf{23.244 \pm 1.553}$ |
| OrganCMNIST | DAW-SI (Ours) | $0.014 \pm 0.008$ | $0.156 \pm 1.424$ | $0.511 \pm 0.140$ | $17.881 \pm 2.624$ |
| | DAWN-SI (Ours) | $\mathbf{0.007 \pm 0.007}$ | $\mathbf{0.151 \pm 1.287}$ | $\mathbf{0.675 \pm 0.116}$ | $\mathbf{21.630 \pm 3.199}$ |

## 3.2 TOMOGRAPHY

We present the results of our methods DAW-SI and DAWN-SI for the tomography task in Table 2 at noise level $p = 5\%$ in the sinogram data for OrganAMNIST and OrganCMNIST datasets (Yang et al., 2023). For this task, DAWN-SI outperformed DAW-SI on all metrics for both datasets. Some example images for the tomography task for different levels of noise are presented in Figures 10 and 11 in Appendix D. In medical image analysis, accurately estimating the size and boundaries of a lobe (such as the one shown in red box in Figure 6) is crucial for diagnostics, especially in cases where its size can influence medical decisions. We quantify uncertainty for both DAW-SI and DAWN-SI by computing mean and standard deviation across 32 samples from the learned posterior. While the mean represents the most probable reconstruction, the standard deviation map highlights regions of higher variability, indicating areas of uncertainty. From Figure 6, DAWN-SI gives a robust understanding of uncertainty by highlighting boundaries of objects as regions of highest uncertainty, a feature similar to what was observed on the MNIST dataset (see Figure 4).

## 4 CONCLUSIONS

In this paper, we presented DAWN-SI, a framework for addressing highly ill-posed inverse problems by efficiently incorporating data and noise into the SI process. Our experiments showed that our proposed method consistently outperformed existing methods in tasks like image deblurring, with significant improvements in key performance metrics. The ability to sample from the learned posterior enables the exploration of the solution space and facilitates in uncertainty quantification, which is critical for real-world applications. Future work will focus on refining the model to enhance efficiency and applicability to more ill-posed problems, including integration of advanced noise modeling techniques for extreme noise conditions.

## ETHICS AND REPRODUCIBILITY STATEMENTS

**Ethics Statement.** In this work, we do not release any datasets or models that could be misused, and we believe our research carries no direct or indirect negative societal implications. We do not work with sensitive or privacy-related data, nor do we develop methods that could be applied to harmful purposes. To the best of our knowledge, this study raises no ethical concerns or risks of negative impact. Additionally, our research does not involve human subjects or crowdsourcing. We also confirm that there are no conflicts of interest or external sponsorships influencing the objectivity or results of this study.

**Reproducibility Statement.** In Section 3, we outline the training methodology employed in our experiments, while in Appendix C, we provide comprehensive supplementary information, including references to the baselines, detailed dataset descriptions, the experimental settings for each task, and the hyperparameter used in our study. All experiments presented in Section 3 were conducted on publicly available benchmarks, while the experiment in Example 2.1 was conducted on simulated data. To further facilitate the reproducibility of our work, we will release all the data and code to reproduce our empirical evaluation upon acceptance.

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

## A  DETAILS ABOUT THE INVERSE PROBLEMS

### A.1  IMAGE DEBLURRING

In this section, we provide technical details about the image deblurring inverse problem used in this work. For this inverse problem, the forward operator $\mathbf{A}$ (a blurring operator) can be represented as a convolution of the input image $I(x, y)$ with a Gaussian Point Spread Function (PSF) kernel, where the kernel is characterized by a standard deviation. The blurring process can be mathematically described as a convolution between a sharp image $I$ and a (Gaussian) blurring kernel $K$, where $I_{\text{blur}}$ is the resultant blurred image,

$$I_{\text{blur}}(x, y) = \mathbf{A}I(x, y) + \epsilon(x, y) = I(x, y) * K(x, y) + \epsilon(x, y), \tag{20}$$

where $I_{\text{blur}}(x, y)$ is the blurred image at pixel location $(x, y)$, $I(x, y)$ is the original sharp image, $K(x, y)$ is the PSF, and $\epsilon(x, y)$ is the additive noise, which is also assumed to be Gaussian with zero mean and a standard deviation equal to $p\%$ of the range of $I(x, y) * K(x, y)$, where $p$ was chosen uniformly between 0% and 20% during training. The Gaussian PSF kernel is given by,

$$K(x, y) = \frac{1}{2\pi\sigma_x\sigma_y} \exp\left(-\frac{x^2}{\sigma_x^2} - \frac{y^2}{\sigma_y^2}\right) \tag{21}$$

All our experiments were run with $\sigma_x = \sigma_y = 3$. The goal of image deblurring is to recover the original sharp image $I(x, y)$ from the blurred image $I_{\text{blur}}(x, y)$. This is typically ill-posed due to the possible presence of noise $\eta(x, y)$ and the loss of high-frequency information caused by the blur.

The convolution operation can be described as a multiplication in the Fourier domain:

$$B(u, v) = I(u, v) \cdot K(u, v) \tag{22}$$

where $I(u, v) = \mathcal{F}\{I(x, y)\}$, $K(u, v) = \mathcal{F}\{K(x, y)\}$ are the Fourier transforms of the sharp image, and the PSF, respectively and $\mathcal{F}$ represents the Fourier transform. The blurred image can then be obtained using the inverse Fourier transform,

$$I_{\text{blur}}(x, y) = \mathcal{F}^{-1}\{B(u, v)\} \tag{23}$$

Moreover, $\mathbf{A}^{\top}$, the adjoint of operator $\mathbf{A}$, which is used to compute the data-embedding $\mathbf{A}^{\top}\mathbf{b}$ for the network, for some data $\mathbf{b}$ is easy to obtain. Since the image blurring operator is symmetric and self-adjoint, we have $\mathbf{A} = \mathbf{A}^{\top}$.

### A.2  APPENDIX - TOMOGRAPHY

In tomography, the goal is to reconstruct the internal image of an object from a series of projections (sinograms) taken at various angles. This process can be described as solving an inverse problem, where the projections are obtained from the original image via the Radon transform $\mathbf{A}$ and inverse Radon transform $\mathbf{A}^{\top}$ is used to reconstruct the original image from the sinogram data.

The forward problem of tomography can be formulated as projecting a 2D image onto a set of 1D sinograms for different angles of view. The mathematical model of the forward projection is expressed as:

$$S = \mathbf{A}I_{\text{pad}} + \epsilon \tag{24}$$

where, $S \in \mathbb{R}^{N_{\text{angles}} \times N_{\text{detectors}}}$ is the sinogram (the set of projections), $I_{\text{pad}} \in \mathbb{R}^{N_{\text{pixels}}}$ is the padded and flattened image of dimension $N_{\text{pixels}}$, $\mathbf{A} \in \mathbb{R}^{N_{\text{angles}} \times N_{\text{detectors}} \times N_{\text{pixels}}}$ is the tomography projection matrix which describes the interaction of each ray with every pixel, and $\epsilon$ is the additive noise term to the data, which was chosen as $p\%$ of the range of values in $\mathbf{A}I_{\text{pad}}$, with $p$ lying between 0% and 20% during training. For our experiments, we set $N_{\text{angles}} = 360$, and $N_{\text{detectors}} = 2s + 1$, where $s \times s$ is the dimension of the original unpadded images.

The image is first padded to account for the fact that projections are taken beyond the boundaries of the object in the image. Using a zero padding of size $s/2$ on all four sides of the image, the padded

version is an image of dimension $2s \times 2s$. The matrix $\mathbf{A}$ is multiplied with the flattened version of the padded image generating the sinogram data.

Similarly, the adjoint $\mathbf{A}^\top$ of the forward operator $\mathbf{A}$, can be used to recover the image, which can be expressed as,

$$I_{\text{pad}} = \mathbf{A}^\top S \tag{25}$$

where, $I_{\text{pad}} \in \mathbb{R}^{N_{\text{pixels}}}$ is the (padded and flattened) reconstructed image, $S \in \mathbb{R}^{N_{\text{angles}} \times N_{\text{detectors}}}$ is the observed sinogram, $\mathbf{A}^\top \in \mathbb{R}^{N_{\text{pixels}} \times N_{\text{angles}} \times N_{\text{detectors}}}$ is the adjoint of the projection matrix. The $I_{\text{pad}}$ can be reshaped and unpadded to obtain an image of dimension $s \times s$.

## B  EVALUATION METRICS

In this section, we provide detailed explanation of the evaluation metrics used in our experiments for 2D images, including the Mean Squared Error (MSE), Misfit, Structural Similarity Index Measure (SSIM), and Peak Signal-to-Noise Ratio (PSNR).

### B.1  MEAN SQUARED ERROR (MSE)

The Mean Squared Error (MSE) measures the average squared difference between the pixel intensities of the original image and the reconstructed image. Given two images $I_{\text{true}}(x, y)$ (the ground truth image) and $I_{\text{rec}}(x, y)$ (the reconstructed image), MSE is calculated as:

$$\text{MSE} = \frac{1}{H \cdot W} \sum_{x=1}^{H} \sum_{y=1}^{W} \left( I_{\text{true}}(x, y) - I_{\text{rec}}(x, y) \right)^2 \tag{26}$$

where:

- $H$ and $W$ are the height and width of the image,
- $I_{\text{true}}(x, y)$ is the pixel value at location $(x, y)$ in the ground truth image,
- $I_{\text{rec}}(x, y)$ is the corresponding pixel value in the reconstructed image.

A lower MSE indicates better reconstruction performance.

### B.2  MISFIT

The Misfit metric measures how well the forward model of the reconstructed image fits the actual observed data. For 2D images, given a forward operator $A$ and observed data $b$, the Misfit is calculated as:

$$\text{Misfit} = \frac{1}{2} \sum_{x=1}^{H} \sum_{y=1}^{W} \left( \mathbf{A} I_{\text{rec}}(x, y) - b(x, y) \right)^2 \tag{27}$$

where:

- $\mathbf{A}$ is the forward operator (e.g., a blurring or projection operator),
- $I_{\text{rec}}(x, y)$ is the reconstructed image,
- $b(x, y)$ is the observed (blurred or noisy) image.

A lower Misfit indicates that the data obtained from the reconstructed image is consistent with the observed data.

### B.3 STRUCTURAL SIMILARITY INDEX MEASURE (SSIM)

The Structural Similarity Index Measure (SSIM) assesses the perceptual similarity between two images by considering luminance, contrast, and structure. For 2D images, SSIM is defined as:

$$\text{SSIM}(I_{\text{true}}, I_{\text{rec}}) = \frac{(2\mu_{\text{true}}\mu_{\text{rec}} + C_1)(2\sigma_{\text{true,rec}} + C_2)}{(\mu_{\text{true}}^2 + \mu_{\text{rec}}^2 + C_1)(\sigma_{\text{true}}^2 + \sigma_{\text{rec}}^2 + C_2)} \tag{28}$$

where:

- $\mu_{\text{true}}$ and $\mu_{\text{rec}}$ are the local means of the ground truth and reconstructed images, respectively,
- $\sigma_{\text{true}}^2$ and $\sigma_{\text{rec}}^2$ are the local variances of the ground truth and reconstructed images,
- $\sigma_{\text{true,rec}}$ is the local covariance between the two images,
- $C_1$ and $C_2$ are constants to avoid division by zero.

SSIM values range from $-1$ to $1$, where $1$ indicates perfect structural similarity.

### B.4 PEAK SIGNAL-TO-NOISE RATIO (PSNR)

The Peak Signal-to-Noise Ratio (PSNR) measures the quality of the reconstructed image compared to the ground truth image. Since our images are normalized between 0 and 1, PSNR is defined as:

$$\text{PSNR} = -10 \cdot \log_{10}(\text{MSE}) \tag{29}$$

wherel, MSE is the Mean Squared Error between the ground truth and reconstructed images. Higher PSNR values indicate better reconstruction quality.

## C EXPERIMENTAL SETTINGS

For the image deblurring task, the experiments were conducted on MNIST, STL10 and CIFAR10 datasets. The networks used were Diffusion model, InverseUNetODE as baselines and our proposed methods DAW-SI and DAWN-SI. The key details of the experimental setup for each of these methods are summarized in Tables 3 to 5. For the tomography task, the experiments were conducted using DAW-SI and DAWN-SI methods on OrganAMNIST and OrganCMNIST datasets derived from the MedMNIST data library (Yang et al., 2023). The key details of the experimental setup are summarized in Table 6. All our experiments were conducted on an NVIDIA A6000 GPU with 48GB of memory. Upon acceptance, we will release our source code, implemented in PyTorch (Paszke et al., 2017).

## D VISUALIZATION

Table 3: Experimental details for the image deblurring task for our methods DAW-SI and DAWN-SI.

| Component | Details |
|---|---|
| Datasets | MNIST, STL10 and CIFAR10. The dimension of images in these datasets were $28 \times 28$, $64 \times 64$, $32 \times 32$, respectively |
| Network architectures | MNIST and CIFAR10:
**DAW-SI**: UNet with 3 levels with [$c$, 16, 32] filters with residual blocks and time and data embedding at each level. Time embed dimension = 256;
**DAWN-SI**: UNet with 3 level levels with [$c$, 16, 32] filters with residual blocks and time, data and noise embedding at each level. Time and noise embed dimension = 256.
Here $c$ = 1 and 3 for MNIST and CIFAR10, respectively.

STL10:
**DAW-SI**: UNet with 5 levels with [3, 16, 32. 64, 128] filters with residual blocks and time and data embedding at each level. Time embed dimension = 256;
**DAWN-SI**: UNet with 5 levels with [3, 16, 32, 64, 128] filters with residual blocks and time, data and noise embedding at each level. Time and noise embed dimension = 256 |
| Number of trainable parameters | MNIST and CIFAR10:
**DAW-SI**: 1,038,398;
**DAWN-SI**: 1,066,497

STL10:
**DAW-SI**: 8,009,824;
**DAWN-SI**: 8,177,955 |
| Loss function | MSE loss for the velocity and misfit terms, Equation (14) |
| Optimizer | Adam (Kingma and Ba, 2014) |
| Learning rate (lr) schedule | CosineAnnealingLR with $\text{lr}_{\text{init}} = 10^{-4}$, $\text{lr}_{\text{min}} = 10^{-6}$ and $T_{\text{max}}$ = max_epochs |
| Stopping criterion | **DAW-SI**: max_epochs = 3000;
**DAWN-SI**: max_epochs = 3000 |
| Integrator for ODE | Fourth-Order Runga-Kutta with step size $h = 1/100$ |

Table 4: Experimental details for the image deblurring task for Diffusion model (Chung et al., 2022a).

| Component | Details |
|---|---|
| Datasets | MNIST, STL10, and CIFAR10. The dimension of images in these datasets were 28x28, 64x64, 32x32, respectively |
| Network architectures | MNIST: UNet with 4 levels with [1, 16, 32, 64] filters and sinusoidal time embedding (1000 time steps) embedded at each level
STL10: UNet with 6 levels with [3, 16, 32, 64, 128, 128] filters and sinusoidal time embedding (1000 time steps) embedded at each level
CIFAR10: UNet with 4 levels with [3, 16, 32, 64] filters and sinusoidal time embedding (1000 time steps) embedded at each level |
| Number of training parameters | MNIST: 1,360,712
STL10: 8,565,086
CIFAR10: 1,365,598 |
| Loss function | MSE loss between $SNR \cdot x_{\text{rec}}$ and $SNR \cdot x_1$, where $SNR$ is the signal-to-noise ratio at each time step, $x_{\text{rec}}$ is the reconstructed image and $x_1$ is the original image |
| Optimizer | Adam (Kingma and Ba, 2014) |
| Learning rate (lr) | $10^{-4}$ (constant) |
| Stopping criterion | max_epochs = 2000 |

Table 5: Experimental details for the image deblurring task for InverseUNetODE (Eliasof et al., 2024).

| Component | Details |
|---|---|
| Datasets | MNIST, STL10, and CIFAR10. The dimension of images in these datasets were 28x28, 64x64, 32x32, respectively |
| Network description | UNet with each level utilizing a combination of convolutional layer embedding for feature extraction and hyperUNet layer for hierarchical feature refinement. The network incorporates the forward problem by applying the adjoint $\mathbf{A}^\top$ of forward problem to the residual at each layer to iteratively correct the estimate of the reconstructed image. |
| Network architectures | MNIST: 3 levels with 3 hidden units per level and 3 nested layers within each hyperUNet layer
STL10: 5 levels with 8 hidden units per level and 3 nested layers within each hyperUNet layer
CIFAR10: 5 levels with 8 hidden units per level and 3 nested layers within each hyperUNet layer |
| Number of training parameters | MNIST: 2,089,887
STL10: 6,503,920
CIFAR10: 6,503,920 |
| Loss function | MSE loss between $x_{\text{recon}}$ (predicted image) and $x_1$ (clean image) |
| Optimizer | Adam (Kingma and Ba, 2014) |
| Learning rate (lr) | $10^{-4}$ (constant) |
| Stopping criterion | max_epochs = 1000 |

Table 6: Experimental details for the tomography task for our methods DAW-SI and DAWN-SI.

| Component | Details |
|---|---|
| Datasets | OrganAMNIST and OrganCMNIST from the MedMNIST dataset (Yang et al., 2023). The dimension of images in both these datasets were $64 \times 64$. |
| Network architectures | **DAW-SI**: UNet with 5 levels with [3, 16, 32. 64, 128] filters with residual blocks and time and data embedding at each level. Time embed dimension = 256;
**DAWN-SI**: UNet with 5 levels with [3, 16, 32, 64, 128] filters with residual blocks and time, data and noise embedding at each level. Time and noise embed dimension = 256 |
| Number of trainable parameters | **DAW-SI**: 8,009,824;
**DAWN-SI**: 8,177,955 |
| Loss function | MSE loss for the velocity and misfit terms, Equation (14) |
| Optimizer | Adam (Kingma and Ba, 2014) |
| Learning rate (lr) schedule | CosineAnnealingLR with $\text{lr}_{\text{init}} = 10^{-4}$, $\text{lr}_{\text{min}} = 10^{-6}$ and $T_{\text{max}}$ = max_epochs |
| Stopping criteria | **DAW-SI**: max_epochs = 3000;
**DAWN-SI**: max_epochs = 3000 |
| Integrator for ODE | Fourth-Order Runga-Kutta with step size $h = 1/100$ |

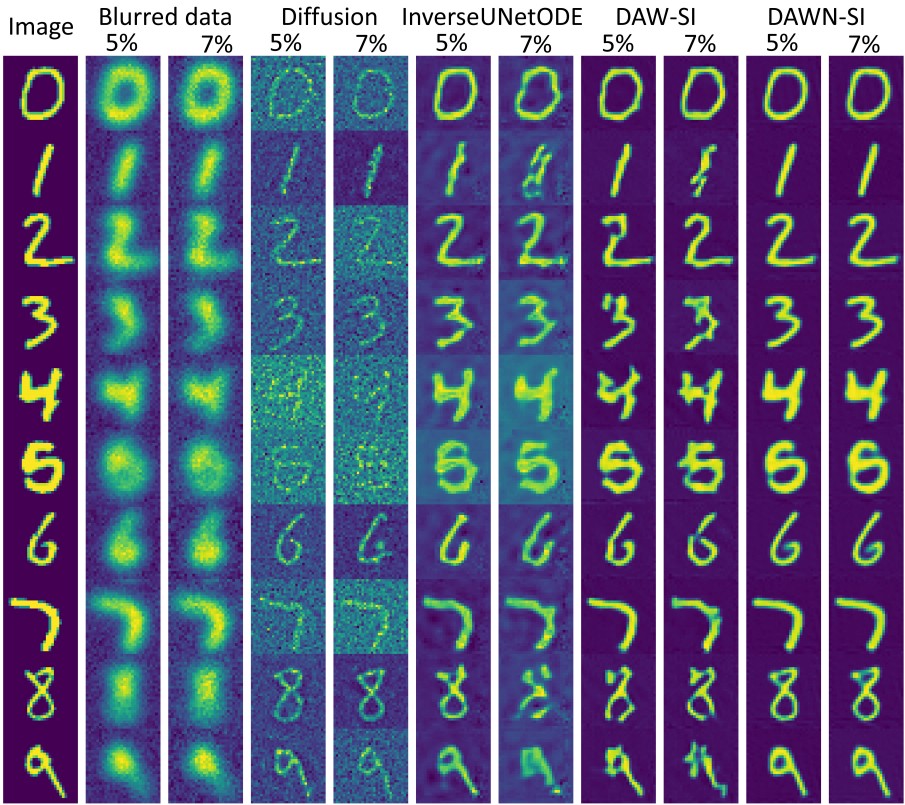

Figure 7: Comparison between different methods for the image deblurring task on the MNIST dataset for noise levels $p = 5\%$ and 7% added to the blurred data.

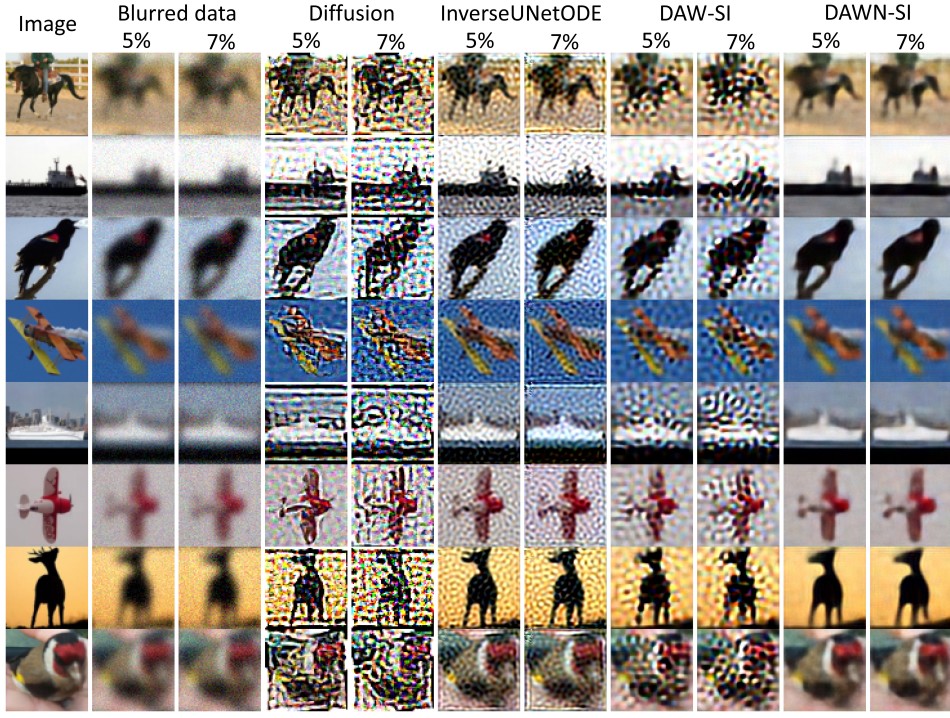

Figure 8: Comparison between different methods for the image deblurring task on the STL10 dataset for noise levels $p = 5\%$ and 7% added to the blurred data.

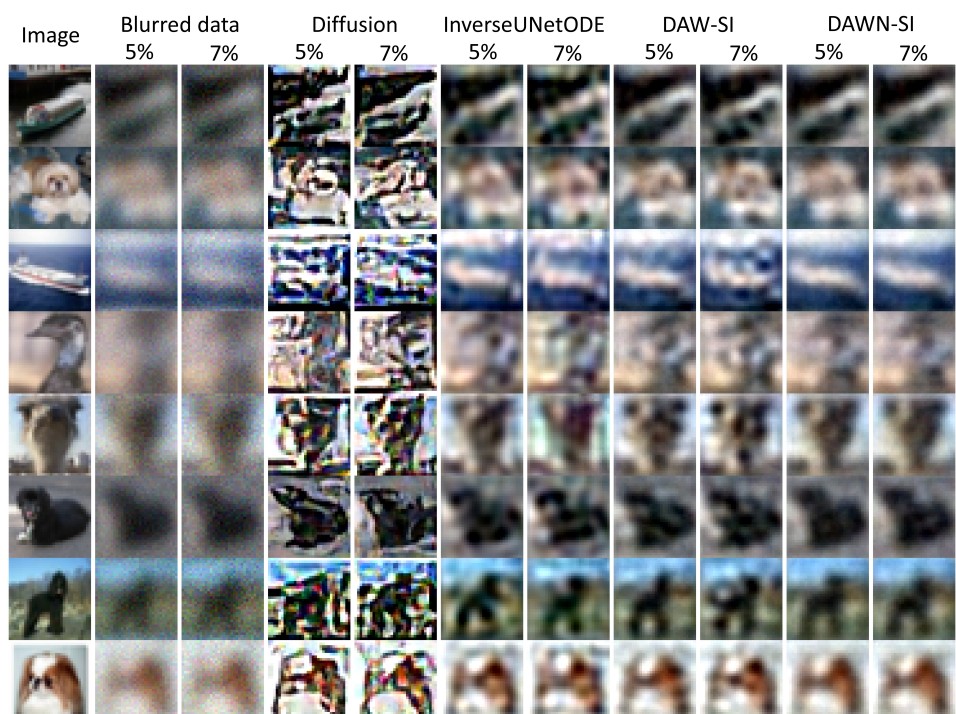

Figure 9: Comparison between different methods for the image deblurring task on the CIFAR10 dataset for noise levels $p = 5\%$ and $7\%$ added to the blurred data.

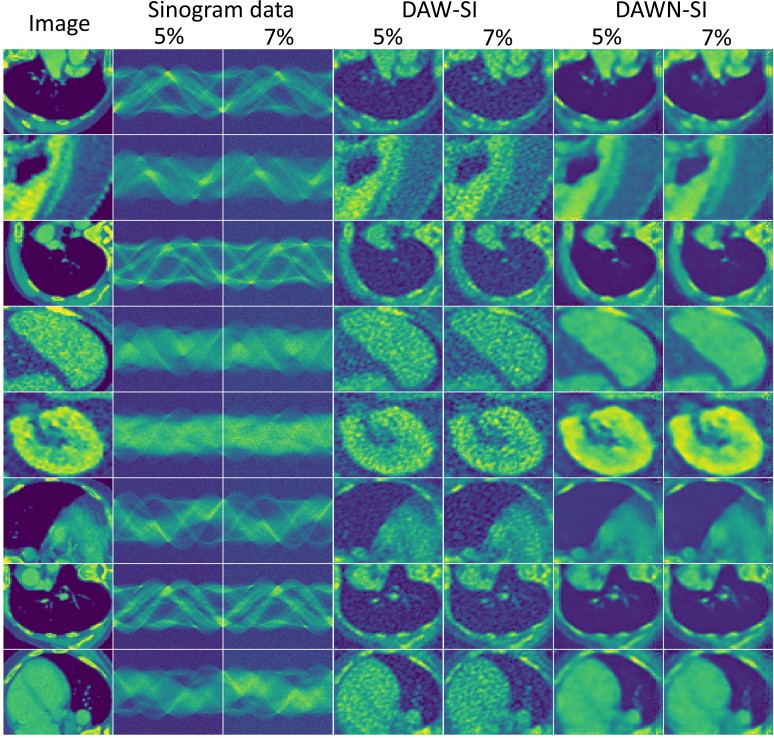

Figure 10: Comparison between DAW-SI and DAWN-SI methods for the tomography task on the OrganAMNIST dataset for noise levels $p = 5\%$ and $7\%$ added to the sinogram data.

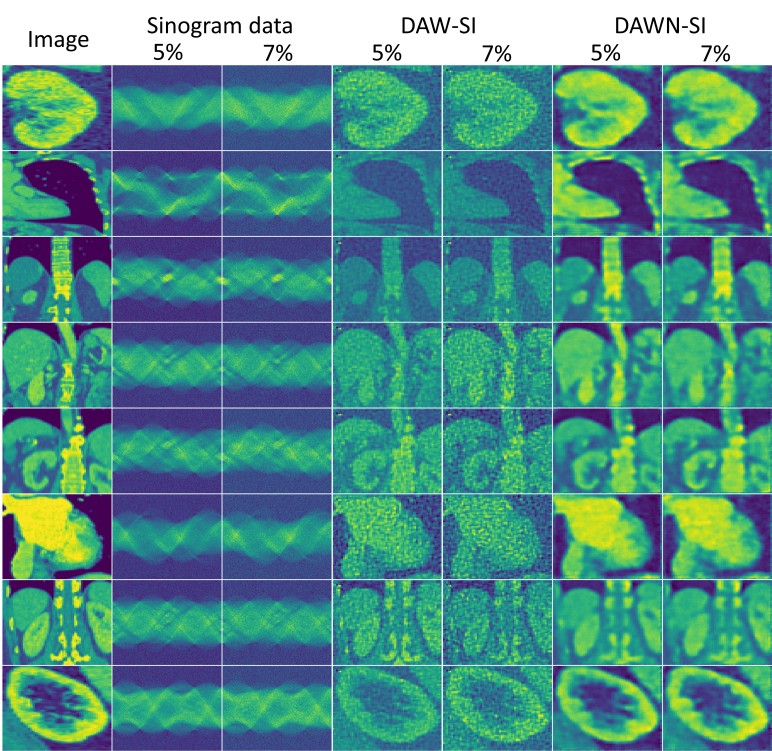

Figure 11: Comparison between DAW-SI and DAWN-SI methods for the tomography task on the OrganCMNIST dataset for noise levels $p = 5\%$ and 7% added to the sinogram data.

