# OpenReview forum: "DAWN-SI: Data-Aware and Noise-Informed Stochastic Interpolation for Solving Inverse Problems"
_ICLR.cc/2025/Conference — ICLR 2025 Conference Withdrawn Submission_

### Official Review · Reviewer_sytH · 2024-10-27

**Soundness:** 2
**Presentation:** 2
**Contribution:** 2
**Rating:** 3
**Confidence:** 4

**Summary:**

This paper proposes a new method for training a denoising diffusion bridge in a data-aware, noise-informed manner to address inverse problems. The data-aware component leverages a deep neural network to learn latent control variables from simulated labels specifically tailored to the inverse problem setting. The noise-informed aspect enables the network to incorporate noise level information from the imaging process itself. However, the overall motivation and empirical effectiveness of this approach are not sufficiently compelling. In its current form, the work feels incomplete, with significant potential for further refinement and improvement.

**Strengths:**

(1) This paper suggests that an additional module can be trained to learn a tailored regularization specific to an inverse problem, which facilitates the use of a denoising diffusion bridge approach for solving it.
(2) This paper demonstrates that informing the network about the noise level in the measurements can enhance model stability compared to methods that rely on blind noise estimation.

**Weaknesses:**

(1) Insufficient Literature Review: The paper’s literature review lacks depth and coverage of relevant prior work. While the proposed method builds on a denoising diffusion bridge or flow matching approach that incorporates forward model and measurement constraints during training, it misses a thorough discussion of similar approaches. The paper should address prior works on diffusion bridges designed for inverse problems, such as I2SB[1] and InDI[2]. Additionally, relevant methods that integrate forward operators into networks during training, including deep unrolling[3], DEQ[4], and recent deep unrolling diffusion models[5], are not discussed. Furthermore, methods that use pre-trained diffusion bridges or restoration networks as priors with forward model constraints during inference, like CDDB[6] and DRP[7], should be included to provide a more comprehensive background.

(2) Unclear and Unconvincing Contributions: The primary contribution of this paper—introducing a data-aware and noise-informed denoising diffusion bridge for inverse problems—is potentially valuable. However, key theoretical aspects, such as the model's assumptions and underlying intuitions, are insufficiently explained. For instance, without the data embedding, the model resembles a denoising ODE aligned with a score-based approach. Yet, with the addition of data embedding, it’s unclear what the model is fundamentally learning—is it refining denoising, or is it addressing a different objective? Additionally, if the goal is to solve inverse problems, why is a denoising ODE formulation selected over a more targeted approach, such as an OT-ODE (I2SB[1] and InDI[2] ) specifically suited for inverse problems? While the approach aims to incorporate noise and data constraints, these theoretical motivations require more clarity. The paper would also benefit from a broader experimental framework—including more baselines and varied settings—to convincingly demonstrate the method’s robustness and comparative advantage.

(3) Limited Experimental Scope and Comparisons:
1) Commonly Used Settings: The paper lacks comparisons in commonly used settings, such as image deblurring with minimal or no noise—scenarios that are crucial for evaluating real-world applicability and understanding when noise level information is more important .
2) Larger Datasets: Experiments on larger image size (at least 256*256) datasets, such as ImageNet, are missing. This is crucial for assessing the method’s efficiency and scalability, particularly as the study presents itself as an empirical investigation.
3) Comprehensive Baselines: The paper does not include a sufficient set of baseline comparisons for image deblurring. Specifically, it should compare against (1) diffusion model-based baselines like DDS [8] and DiffPIR [9]; (2) diffusion bridge-based methods such as I2SB[1] and InDI[2] and CDDB[6]; and (3) deep unrolling methods like USRNet [10].

(4) Lack of Baseline Comparison for CT Reconstruction: For CT reconstruction, there is no comparison against existing deep unrolling[11] or diffusion model-based methods[12], despite the availability of many relevant approaches for this task. The absence of these comparisons limits the comprehensiveness of the evaluation.

(5) Unconvincing Performance: The performance are not convincing. For example, the uncertainty maps in Figure 4 provide limited interpretable information, and the reconstruction quality lacks sufficient visual clarity. While these results may be adequate for downstream tasks like classification, additional experiments and comparative results with relevant baselines are needed to substantiate the model's effectiveness.


Reference:
 [1] G. Liu, A. Vahdat, D. Huang, E. A Theodorou, W. Nie, and A. Anandkumar. I2sb: image-to-image schr¨odinger bridge. In Proc. Int. Conf. Machine Learning (ICML), pp. 22042–22062, 2023.
[2] M. Delbracio and P. Milanfar. Inversion by direct iteration: An alternative to denoising diffusion for image restoration. Trans. on Mach. Learn. Research, 2023. ISSN 2835-8856.
[3] J. Zhang and B. Ghanem. ISTA-Net: Interpretable optimization-inspired deep network for image compressive sensing. In Proc. IEEE Conf. Comput. Vis. Pattern Recognit. (CVPR), pp. 1828–1837, 2018.
[4] D. Gilton, G. Ongie, and R. Willett. Deep equilibrium architectures for inverse problems in imaging. IEEE Trans. Comput. Imag., 7:1123–1133, 2021a.
[5] Guo, L. and Wang, C. and Yang, W. and Huang, S. and Wang, Y. and Pfister, H. and Wen, B. "Shadowdiffusion: When degradation prior meets diffusion model for shadow removal." In Proceedings of the IEEE/CVF Conference on Computer Vision and Pattern Recognition, pp. 14049-14058. 2023.
[6] H. Chung, J. Kim, and J. C. Ye, “Direct diffusion bridge using data consistency for inverse problems,” Advances in Neural Information
Processing Systems, vol. 36, 2024.
[7] Y. Hu, M. Delbracio, P. Milanfar, and U. S. Kamilov, “A Restoration Network as an Implicit Prior,” Proc. Int. Conf. Learn. Represent. (ICLR 2024) (Vienna, Austria, May 7-11).
[8] H. Chung, S. Lee, and J. C. Ye. Decomposed diffusion sampler for accelerating large-scale inverse problems. In Proc. Int. Conf. on Learn. Represent. (ICLR), 2024.
[9] Y. Zhu, K. Zhang, J. Liang, J. Cao, B. Wen, R. Timofte, and L. Van G. Denoising diffusion models for plug-and-play image restoration. In Proc. IEEE Conf. Comput. Vis. and Pattern Recognit. (CVPR), pp. 1219–1229, 2023.
[10] K. Zhang, W. Zuo, and L. Zhang. Deep plug-and-play super-resolution for arbitrary blur kernels. In Proc.
IEEE Conf. Comput. Vis. Pattern Recognit. (CVPR), pp. 1671–1681, Long Beach, CA, USA, June 16-20, 2019.
[11] D. Hu, Y. Zhang, J. Liu, S. Luo, and Y. Chen. Dior: deep iterative optimization-based residual-learning for limited-angle ct
reconstruction. IEEE Trans. on Med. Imag., 41(7):1778–1790, 2022. 2, 5.
[12] J. Liu, R. Anirudh, J. J. Thiagarajan, S. He, K. A. Mohan, U. S. Kamilov, and H. Kim, “DOLCE: A Model-Based Probabilistic Diffusion Framework for Limited-Angle CT Reconstruction,” Proc. IEEE Int. Conf. Comp. Vis. (ICCV 2023) (Paris, France, October 2–6), pp. 10498-10508.

**Questions:**

For DAWN-SI, the model requires the input of the noise level in the measurement during inference. How would this noise level be obtained in real-world scenarios? Additionally, how sensitive is the model to inaccuracies in the estimated noise level?

---

### Official Review · Reviewer_k7cK · 2024-11-03

**Soundness:** 1
**Presentation:** 2
**Contribution:** 1
**Rating:** 3
**Confidence:** 5

**Summary:**

The paper presents DAWN-SI, a framework that employs Data-Aware and Noise-Informed Stochastic Interpolation for solving ill-posed inverse problems, specifically in the contexts of image deblurring and tomography. The authors argue that DAWN-SI enhances the robustness of inverse problem solutions by embedding both data characteristics and noise information directly into the model’s interpolation process. The approach is tested against existing methods, including diffusion models and InverseUNetODE, with results indicating improved performance in handling varying noise levels. Additionally, DAWN-SI allows for uncertainty quantification by generating multiple plausible solutions, a feature the authors emphasize as valuable for practical applications in fields requiring stable inverse solutions.

**Strengths:**

The DAWN-SI framework has a few positive aspects. Its integration of noise characteristics into the interpolation process is intended to make it adaptable to noisy or incomplete data. Additionally, its capacity to generate multiple plausible solutions allows for uncertainty quantification, which could be useful in applications requiring an understanding of solution variability. Finally, DAWN-SI’s broad framing for inverse problems suggests potential applications beyond image deblurring and tomography, possibly extending to areas like medical imaging.

**Weaknesses:**

**Concerns with the Choice of Adjoint Operator.** The selection of the adjoint operator $A^T$ is unconvincing in this context. Although $A^T$ plays a central role in the proposed algorithm, it is not clear that this choice is optimal. Alternative approaches, such as a direct mapping learned through a neural network, could be more effective. More critically, I am skeptical that the adjoint operator is appropriate for the tomography problem, where there is a clear distinction between the reconstruction and measurement domains. In deblurring, the adjoint operator, due to the properties of the blurring kernel, can serve as a good proxy for the inverse operator, which likely explains the satisfactory results in Figure 4. However, the tomography results shown in Figures 11 and 12 are less convincing, reflecting potential limitations of the adjoint approach in this domain. This observation leads to further concerns below.

**Weak Baseline Comparisons for CT.** The results for CT reconstruction lack comprehensive baseline comparisons, making it difficult to evaluate the effectiveness of the proposed method. Table 2 presents the only statistical results for CT, yet without any standard baseline methods for comparison, the quality of these results is challenging to assess. Furthermore, the visual results in Figures 11 and 12 appear underwhelming, raising doubts about the method’s practical effectiveness in CT applications.


**Questionable Applicability to Nonlinear Inverse Problems.** The authors suggest that their approach can extend to nonlinear inverse problems, but this claim is debatable. While it may be theoretically possible, finding a suitable transformation from the data domain to the reconstruction domain is exceptionally challenging in practice for nonlinear problems. If such a transformation were already identified, it would solve a significant portion of the problem, as this is often the most difficult aspect of nonlinear inversion. Moreover, established methods like the adjoint-state approach, which are sometimes adapted for nonlinear problems, tend to be computationally intensive and may not be feasible in practical settings.

**Questions:**

1. Could the authors clarify whether the adjoint operator was selected based on theoretical considerations or empirical results specific to this problem?

2. Have the authors explored other operators or transformations to improve reconstruction accuracy, particularly for CT applications?

3. Why were no standard baseline methods included for comparison in the CT results, given that such comparisons are critical for assessing model performance?

4. Could the authors explain why the CT results in Table 2 lack statistical comparisons with established methods like plug and play priors method?

5. Could the authors clarify the extent to which their approach has been tested on nonlinear inverse problems, if at all?

6. Are there specific types of nonlinear inverse problems that the authors envision their method could handle effectively?

---

### Official Review · Reviewer_DBiu · 2024-11-04

**Soundness:** 2
**Presentation:** 3
**Contribution:** 2
**Rating:** 6
**Confidence:** 3

**Summary:**

The authors propose a method for solving linear inverse problems using stochastic interpolation framework. They motivate their method by suggesting that it is better suited for highly ill-posed inverse problems where either the measurement matrix is very low rank or the measurement noise is very high. They describe a fidelity term in the loss function and they describe the details of the architecture. Finally they show numerical results for de-blurring and tomography.

**Strengths:**

1. Paper is written very clearly and is easy to read.
2. Presenting a toy example and the architecture diagram is useful.

**Weaknesses:**

1. A major limitation of this method is that it is specific to a particular inverse problem tied to a forward process, A. Since the measured data from A is included in the training procedure, the network can be used only on the specific problem it is trained on. So, this method loses a big advantage that is the main motivation for using diffusion or interpolant networks in solving inverse problems: namely their generality (one network can be used to solve several problems). This limitation needs to be discussed upfront. Also, this limitation raises a question: what is the motivation to do this? If you lose the generality of the method, then why not train a network to solve the inverse problem directly using a supervised scheme? What is the point of all of this apparatus?
2. The method, as presented, is limited to linear inverse problems. This needs to be reflected early on, in the title or abstract.
3. The paper proposes that computing uncertainty is one of the contributions. This however has been explored previously in the context of inverse problems using diffusion models. [1] is one example. Authors should consider adjusting the text to include previous work on this topic.
4. The idea of averaging across samples to get the mean of the posterior was proposed in early works [2] on solving inverse problems using diffusion models. Again, authors should include references to prior related work.


[1] Nehme, Elias, Rotem Mulayoff, and Tomer Michaeli. "Hierarchical Uncertainty Exploration via Feedforward Posterior Trees." arXiv preprint arXiv:2405.15719 (2024).

[2] Kadkhodaie, Zahra, and Eero Simoncelli. "Stochastic solutions for linear inverse problems using the prior implicit in a denoiser." Advances in Neural Information Processing Systems 34 (2021): 13242-13254.

**Questions:**

1. The reasoning at lines 195 to 199 seems erroneous to me. The fidelity of the solution to the measurement does not depend on whether you use posterior directly or use its factorization into prior and likelihood. Instead, the fidelity relies on two other factors: first, dimensionality of the measurement process (lower rank A results in heavier reliance on the prior, hence less fidelity to original image), and second, the loss function used to get the best estimate. For example using samples results in low fidelity, but MMSE results in higher fidelity.
2. In table 1, is the results presented for diffusion model coming from indivudual samples or average across 32 samples? It is only fair to compare to the average since for DAWN method averages are presented. Obviously the average results in lower MSE because it approximate posterior mean which is the minimizer of MSE.
3. The image deblurring results presented in figure 4 seem low quality. This is concerning because deblurring without noise  (which seems to be the case in figure 4) is a very simple problem. In fact, it has an analytical exact solution: multiplying by the inverse of the blurring kernel in the Fourier domain. I believe the solutions for debluring with noise using deep nets is way better than what's presented here.

---

### Official Review · Reviewer_rPYm · 2024-11-04

**Soundness:** 2
**Presentation:** 2
**Contribution:** 2
**Rating:** 5
**Confidence:** 4

**Summary:**

The paper proposes to use conditional flow matching models as inverse problem solvers to tackle highly ill-posed inverse problems. The proposed velocity model is conditioned on both the measurements and the level of measurement noise to enhance robustness. The experiments cover different datasets and the results compared to DPS and InverseUNetODE show favorable performance in distortion.

**Strengths:**

* The paper is written clearly, and the math is rigorous and easy to follow.
* The results show that in distortion metrics the approximated mean does better than the baselines (especially DPS).

**Weaknesses:**

* The idea of training conditional flow matching models for inverse problem-solving is not new. See \[1\] for example:
  \[1\] Zhu et al., “**FlowIE: Efficient Image Enhancement via Rectified Flow**”, CVPR 2024\.
* Important baselines and related work seem to be omitted from the comparisons. For example, in deblurring, DDRM/DDNM \[2\]-\[3\] pose natural baselines that can compete with DAWN-SI while efficiently handling the operator's null space mentioned in L180-190.
  \[2\] Kawar et al., “**Denoising Diffusion Restoration Models**”, NeurIPS 2022\.
  \[3\] Wang et al., “**Zero-shot Image Restoration Using Denoising Diffusion Nullspace Model**”, ICLR 2023\.
* The experiments section mostly covers simple datasets, with no application to standard image restoration datasets such DIV2K, Flicker2K, Kodak, BSD, etc. To enable proper benchmarking of DAWN-SI advantages it makes sense to apply it to more standard natural image datasets.
* The quality of the derived uncertainty visualization is questionable. The diversity of the resulting posterior samples is not showcased for the most part except for Fig. 6\. For example, in Figure 4 (other than the MNIST dataset), the uncertainty heatmaps offer very little insight into the possible set of solutions.

My decision is mainly due to omitted related background, lack of proper benchmarking, and the underwhelming results in the experiments section.

Minor typos/issues:

* L244 \- why did you use italics for the entire toy example?
* Eq. (18) is missing a square on $\\sigma\_{\\bar{\\mathbf{x}}\_1}$.

**Questions:**

* In example 2.1 (and Fig. 2\) it is worth adding more details. Specifically, what is the considered measurement in Fig. 2 right? What is the legend for the color of the points?
* Did you try using a posterior summarization technique that does not ignore pixel correlations to visualize uncertainty? For example, PCA/$K$-means clustering applied to the 32 posterior samples in Fig. 4?
* Can you elaborate on antithetic sampling (Eq. (19))? Is this standard practice? Did you try ablating the effect of this technique on your method? For example, did you measure convergence quality/time with/without it?
* You mention in L422 that the encoding of the backprojected measurement is done with a single convolutional layer. What is the intuition for using such a shallow encoder? Did you try using something more elaborate (e.g. multiple conv layers/Fully connected)?
* In Fig. 6, the samples of DAW-SI seem extremely noisy. Is there an obvious reason for this? Also, the samples of DAWN-SI seem to suffer from a gridding artifact that is later flagged in the uncertainty heatmap obscuring the uncertainty in the highlighted red box. What is the reason for this?
* How do you think your performance will compare against zero-shot flow methods (e.g. [4]) in linear inverse problems such as deblurring?

   \[4\] Pokle et al., “**Training-free linear image inverses via flows**”, arxiv 2023\.

---

### Note · Authors · 2024-12-07

I have read and agree with the venue's withdrawal policy on behalf of myself and my co-authors.